# Identification of Differential Circular RNA Expression Profiles and Functional Networks in Human Macrophages Induced by Virulent and Avirulent *Mycobacterium tuberculosis* Strains

**DOI:** 10.3390/ijms242417561

**Published:** 2023-12-16

**Authors:** Yifan Zhu, Delai Kong, Zijian Wang, Ting Li, Tian Tang, Yongchong Peng, Changmin Hu, Jin Chao, Huanchun Chen, Yingyu Chen, Aizhen Guo

**Affiliations:** 1State Key Laboratory of Agricultural Microbiology, Hubei Hongshan Laboratory, College of Veterinary Medicine, Huazhong Agricultural University, Wuhan 430070, China; zhuyifan@webmail.hzau.edu.cn (Y.Z.); litingtsajlt@163.com (T.L.);; 2National Animal Tuberculosis Para-Reference Laboratory (Wuhan) of Ministry of Agriculture and Rural Affairs, International Research Center for Animal Disease, Ministry of Science and Technology, Huazhong Agricultural University, Wuhan 430070, China; 3Hubei Key Laboratory of Tumor Microenvironment and Immunotherapy, Institute of Infection and Inflammation, Medical College, China Three Gorges University, Yichang 443002, China

**Keywords:** *Mycobacterium tuberculosis*, macrophages, circRNAs, expression profiles, subnetwork

## Abstract

Circular RNAs (circRNAs) are noncoding RNAs with diverse functions. However, most *Mycobacterium tuberculosis* (*M.tb*)-related circRNAs remain undiscovered. In this study, we infected THP-1 cells with virulent and avirulent *M.tb* strains and then sequenced the cellular circRNAs. Bioinformatic analysis predicted 58,009 circRNAs in all the cells. In total, 2035 differentially expressed circRNAs were identified between the *M.tb*-infected and uninfected THP-1 cells and 1258 circRNAs were identified in the virulent and avirulent *M.tb* strains. Further, the top 10 circRNAs were confirmed by Sanger sequencing, among which four circRNAs, namely *circSOD2*, *circCHSY1*, *circTNFRSF21*, and *circDHTKD1*, which were highly differentially expressed in infected cells compared with those in uninfected cells, were further confirmed by ring formation, specific primers, and RNase R digestion. Next, circRNA-miRNA-mRNA subnetworks were constructed, such as *circDHTKD1*/*miR-660-3p*/*IL-12B* axis. Some of the individual downstream genes, such as *miR-660-3p* and *IL-12B*, were previously reported to be associated with cellular defense against pathological processes induced by *M.tb* infection. Because macrophages are important immune cells and the major host cells of *M.tb,* these findings provide novel ideas for exploring the *M.tb* pathogenesis and host defense by focusing on the regulation of circRNAs during *M.tb* infection.

## 1. Introduction

Tuberculosis (TB) is a chronic infectious disease caused by *Mycobacterium tuberculosis* (*M.tb*) infection and remains a leading cause of morbidity and mortality worldwide. In 2022, TB infected 10.6 million people and killed 1.3 million of them [1]. *M.tb* is transmitted from an individual to another via aerosol [2]. Upon inhalation, the droplet nuclei carrying the bacteria enters the lung and the infection is established in the alveolar space [3]. Once in the alveoli, the *M.tb* interacts with lung-resident macrophages, and the macrophages activate and recruit other immune cells, including lung epithelial cells, neutrophils and dendritic cells, to limit the infection [4,5,6]. Macrophages are the primary host cells for *M.tb* infection and persistence, and undergo extensive transcriptional changes in response to *M. tb* infection [7]. These changes involve both coding and noncoding RNAs that regulate host resistance and immunity or pathological processes. Nevertheless, there is no clear understanding of the molecular mechanisms of noncoding RNAs and their interactions between coding and noncoding RNAs that govern the macrophage-*M.tb* interface.

Circular RNA (circRNA) is a class of noncoding RNA (ncRNA) that forms covalently closed circular structures from the 3′ and 5′ ends of pre-mRNA through back-splicing during mRNA processing although some recent evidence has revealed that some circRNAs can be translated into peptides [8,9,10]. CircRNAs can be classified into six types, viz., exon–exon, intron–exon & exon–intron, intron–intron, intergenic–intergenic, exon–intergenic & intergenic–exon, and intergenic–intron & intron–intergenic circRNAs [9,11,12]. Exon circRNAs are the most abundant, and derived from one or more exons of protein-coding genes [13,14]. Intron circRNAs are composed of introns [15]. Exon–intron circRNAs contain both exon and intron sequences because of incomplete splicing or debranching [16,17]. Previous research has demonstrated that circRNAs exhibit diverse functions, such as scaffolding proteins, recruiting other RNAs, and regulating the expression or degradation of specific mRNAs by binding to miRNAs [11].

To date, approximately 40 circRNAs have been reported to be possibly associated with *M.tb* infection or tuberculosis [18]; however, only a few have been preliminarily confirmed. For instance, *circPWWP2A* has been implicated in the death of macrophages infected by *M.tb* by elevating *miR-579* levels [19]. *circ0045474* was found to trigger macrophage autophagy in TB through the *miR-582-5p*/*TNKS2* pathway [20]. *circ0003528* was reported to enhance the polarization of macrophages associated with active pulmonary TB by increasing *CTLA4* expression [21]. Another study has reported that *circTRAPPC6B* inhibited the growth of *M.tb* inside macrophages and induced autophagy by targeting *miR-874-3p* [22]. Moreover, because circRNAs are extremely stable and resistant to degradation because of their covalent bonds [23], they might enable the development of novel diagnostic and treatment methods for TB [24,25,26].

In this study, our aim was to systematically identify and characterize circRNAs that have potential functions associated with *M.tb* infection in the macrophages using virulent and avirulent strains to infect THP-1 cells and then conduct the circRNA sequencing of infected and uninfected THP-1 cells. Our analysis revealed a unique set of circRNAs specific to *M.tb*-infected macrophages and confirmed some of them. We further constructed the circRNA-miRNA-mRNA interaction subnetwork of four novel circRNAs—*circSOD2*, *circCHSY1*, *circTNFRSF21*, and *circDHTKD1*. Our data provide novel ideas about investigating *M.tb* pathogenesis and host defense by focusing on the function of circRNAs during *M.tb* infection.

## 2. Results

### 2.1. CircRNA Profiling in All THP-1 Cells

We used the DCC software (version 0.5.0) to predict the presence of circRNAs from the sequencing data, which generated 58,009 circRNAs corresponding to 8408 parental genes that are distributed throughout all 23 chromosomes in human beings (Figure 1A,B). Among them, only 45.02% of circRNAs were annotated, whereas 54.98% of them were novel according to the circBase database (Figure 1A). All 58,009 circRNAs were classified into six types according to their start and end regions in the genome, wherein most of them (92.53%) were exon-associated, followed by intron-associated (14.68%) and intergenic-associated (5.11%) (Figure 1C). The length of most circRNAs was distributed in the range of 150–2000 nt, with an average length of approximately 450 nt (Figure 1D). From the aspect of abundance, 55.6% of circRNAs had at least one back-splicing read stably in each *M.tb*-infected group (Figure 1E). In all the 8408 parental genes, 52.64% produced fewer than three circRNAs, whereas 17.51% generated >10 circRNAs (Figure 1F).

### 2.2. Analysis of Differentially Expressed circRNAs

To compare the differentially expressed circRNAs in virulent *M.tb* 1458 (virulent *M.tb*)-infected and uninfected THP-1 cells identified 799 differentially expressed circRNAs with 401 upregulated, and 398 downregulated (Figure 2A,D). For avirulent *M.tb* H37Ra (H37Ra)-infected and uninfected THP-1 cells, 1560 circRNAs were differentially expressed, of which 382 were upregulated and 1178 were downregulated (Figure 2B,E). Moreover, 1258 circRNAs were differentially expressed between virulent *M.tb-* and H37R- infected THP-1 cells, with 998 being upregulated and 260 being downregulated (Figure 2C,F). Furthermore, 324 differentially circRNAs were commonly expressed in both virulent *M.tb*- and H37Ra-infected cells compared with those in uninfected cells, with 179 being upregulated and 145 being downregulated (Figure 2G–I). Infection with the avirulent H37Ra resulted in more downregulation of circRNAs than infection with the virulent *M.tb* strain.

Regarding the length of the spliced circRNA sequences, the average length of the downregulated circRNAs in THP-1 cells infected with both virulent and avirulent strains was much longer than that of the upregulated circRNAs compared with that in uninfected THP-1 cells (*p* < 0.001) (Figure 2J,K). However, no significant difference was found between the virulent *M.tb*- and H37Ra-infected THP-1 cells (*p* > 0.05) (Figure 2L).

### 2.3. KEGG and Reactome Pathway Analysis of Differentially Expressed circRNA Parental Genes

To further investigate the potential functional roles of the significantly differentially expressed circRNAs in *M.tb*-infected THP-1 cells, the corresponding genes were subjected to KEGG and Reactome pathway analysis. The KEGG pathway analysis of the parental genes in virulent *M.tb*-infected macrophages revealed the following top four significantly enriched pathways: influenza A, autophagy, cellular senescence, and NF−κB signaling pathways. The top 15 enriched pathways are displayed in a scatter diagram (Figure 3A). Meanwhile, analysis of the H37Ra-infected group revealed the top 15 enriched pathways, including endocytosis, ubiquitin-mediated proteolysis, autophagy, and TGF−β signaling pathways (Figure 3B). For the parental genes corresponding to the differentially expressed circRNAs between virulent *M.tb*- and H37Ra-infected THP-1 cells, the top 15 enriched pathways included endocytosis, proteoglycans in cancer, regulation of actin cytoskeleton, and human T−cell leukemia virus 1 infection (Figure 3C).

We also analyzed the parental genes using Reactome. As depicted in Figure 3D, most parental genes of the differentially expressed circRNAs were related to processes of cell cycle, cell death, and interferon pathways in virulent *M.tb*-infected macrophages compared with those in uninfected cells. In the H37Ra-infected macrophages, most genes were involved in sumoylation and cell cycle (Figure 3E). Furthermore, for the parental genes corresponding to the differentially expressed circRNAs between virulent *M.tb*- and H37Ra-infected THP-1 cells, most genes were involved in DNA repair, organelle biogenesis and maintenance, and cilium assembly (Figure 3F).

### 2.4. Detection of Differentially Expressed circRNAs

Based on fold changes, *p* values, and the abundance in macrophages, we selected the top 20 upregulated and 20 downregulated circRNAs in virulent *M.tb*- or H37Ra-infected macrophages compared with that in controls, which are respectively shown in Figure 4A and Figure 4B.

We used specific primers and performed Sanger sequencing to screen the top 10 differentially expressed circRNAs in virulent *M.tb*- or H37Ra-infected macrophages, and found that the back-splicing sites of *circCHSY1*, *circSOD2*, *circTNFRSF21*, and *circDHTKD1* were confirmed by Sanger sequencing as depicted in Figure 4C–F, which also presents the schematic illustration of these four circRNAs and the details for primer designing (Figure 4C–F).

We next performed RT-PCR using convergent and divergent primers in cDNA and gDNA samples to further validate the existence of these four circRNAs. Our results demonstrated that these circRNAs could be amplified only in cDNA, but not in gDNA, indicating that they were formed at the posttranscriptional level. Furthermore, the circular features of these circRNAs were confirmed using the RNase R digestion experiment, which revealed that all the circRNAs were resistant to RNase R treatment in THP-1 cells, whereas the expression level of their linear mRNA counterparts was significantly decreased by RNase R digestion (Figure 4C–F). These data suggested the presence of these four circRNAs in *M.tb*-infected THP-1 cells.

To further validate the sequencing results, we conducted RT-qPCR using the same samples as in sequencing. Consistent with the sequencing results, the expressions of *circSOD2* (Figure 5A) and *circCHSY1* (Figure 5B) were significantly upregulated (*p* < 0.01) in virulent *M.tb*- and H37Ra-infected THP-1 cells, whereas the expressions of *circTNFRSF21* (Figure 5C) and *circDHTKD1* (Figure 5D) were significantly downregulated (*p* < 0.05). These findings suggested that these four circRNAs had a similar expression pattern in THP-1 cells infected with either virulent *M.tb* or avirulent H37Ra, further confirming the high reliability of this circRNA profile (Figure 5).

### 2.5. Construction of M.tb Infection-Related circRNA-miRNA-mRNA Competing Endogenous Interaction Network

CircRNAs in the cytoplasm can act as super sponges to modulate the expression of miRNA targets in tumor or TB regulation [22,27,28]. To detect the subcellular localization of *circSOD2*, *circCHSY1*, *circTNFRSF21*, and *circDHTKD1*, we performed nucleocytoplasmic isolation qPCR, which revealed that all the *circSOD2*, *circCHSY1*, *circTNFRSF21*, and *circDHTKD1* (Figure 6A,B) were highly expressed in the cytoplasm compared with those in the nucleus in *M.tb*-infected THP-1 cells.

Next, to elucidate the potential role of these four highly expressed circRNAs in *M.tb*-infected THP-1 cells, we constructed an *M.tb* infection circRNA-miRNA-mRNA competing endogenous RNA (ceRNA) network based on these validated circRNAs and bioinformatic predictions. We first used miRanda to estimate the potential binding sites between the selected circRNAs and their target miRNAs. We selected miRNAs that met the criterion maximum energy ≤ −20 and score >140. Then, for each of these four circRNAs, we used miRwalk, miRDB, and TargetScan to predict the targets of all the top 10 miRNAs obtained in this manner. We selected the predicted targets that were supported by at least two softwares for further analysis. Moreover, to determine the relationship between these target genes and *M.tb* infection, we matched all the miRNA targets to the tuberculosis signaling pathway ([*hsa-05152*]). We found that 47 target genes were enriched in the tuberculosis signaling pathway and might have potential associations with *M.tb* infection (Appendix A). Finally, we constructed a circRNA-miRNA-mRNA interaction network based on all the above-described results to further explore the underlying mechanism of circRNA regulation in *M.tb* infection (Figure 6C).

## 3. Discussion

We used virulent and avirulent *M.tb* strains to infect THP-1 cells and identified 58,009 circRNAs generated by 8408 parental genes, among which 2035 differentially expressed circRNAs were identified. Moreover, infection with the avirulent H37Ra resulted in more downregulation of circRNAs than infection with the virulent *M.tb* strain. Meanwhile, the exonic circRNAs were the most abundant type, and the majority of circRNAs had a length of 150–2000 nt, with an average length of approximately 450 nt. These findings were consistent with the general characteristics of circRNAs reported in previous studies [29,30]. To our best knowledge, this is the first study on the expression profiles of circRNAs in macrophages infected with virulent and avirulent *M.tb* strains.

### 3.1. CircRNAs Expression in Macrophages

Among the identified 58,009 circRNAs, most had low expression levels in macrophages, whereas a small proportion of circRNAs had much higher expression levels than others, which is consistent with previous studies on circRNAs in human cells [31,32,33,34,35]. Due to the specific structure of circRNAs, which makes them more resistant to RNase degradation, these highly abundant circRNAs would be significant to help understand the mechanism of circRNA formation and stability and design synthetic circRNAs as drug delivery tools. Nonetheless, future studies are required to disclose the structural characteristics of these circRNAs.

In contrast, one parental gene can develop more circRNAs [8,9,10]. In the present study, 52.64% of the parental genes (total 8408) produced 1–3 circRNAs, whereas 17.51% produced >10 circRNAs, indicating that the parental genes were extensively back-spliced. Most circRNAs are exonic circRNAs, which might compete with their parental linear transcripts and affect their coding potential. It has also been demonstrated that exon-derived circRNAs can interact with U1 and SNPs, which can also affect the transcription of their parental genes [36]. This suggests that the function of their linear transcripts is significantly altered when extensive back-splicing occurs on one gene. These findings enrich the current knowledge on the generation of circRNAs from their parental genes.

### 3.2. Differential circRNAs Associated with Virulent and Avirulent M.tb

Because different circRNA profiles induced by *M.tb* strains with different virulence patterns might reveal a potential mechanism of pathogenesis related to *M.tb* infection and tuberculosis, we compared the differential expression of circRNAs after infection with virulent or avirulent *M.tb* strains. Based on the uninfected control, infection with the virulent and avirulent *M.tb* strains commonly induced 324 differentially circRNAs, which might be related to mycobacterial infection unrelated to *M.tb* virulence. However, 475 and 1236 unique differentially expressed circRNAs were respectively identified in virulent *M.tb*- and H37Ra-infected THP-1 cell groups, which might be associated with *M.tb* virulence, helping to elucidate *M.tb* pathogenesis, and be used as diagnostic target candidates for tuberculosis [37,38,39].

To date, approximately 40 circRNAs were considered to have potential functions in tuberculosis, of which 13 were demonstrated to affect the interaction between *M.tb* and host cells [18,19,20,21,22,40]. This indicated that most of our circRNAs related to virulent and avirulent *M.tb* infection are novel and need to be further investigated.

In the present study, we selected the top 10 upregulated and downregulated circRNAs common for virulent and avirulent *M.tb* infection for further confirmation and demonstrated the presence of four circRNAs, viz., *circSOD2*, *circCHSY1*, *circTNFRSF21*, and *circDHTKD1*, using divergent and convergent primers to perform RNase R digestion, amplification of cDNA and gDNA, and Sanger sequencing. Based on the circRNA database, we discovered that *circSOD2* was back-spliced by exons 4 and 5 of *SOD2* transcripts, *circCHSY1* was back-spliced by only exon 2 of *CHSY1* transcripts, *circTNFRSF21* was back-spliced by exons 5 and 6 of *TNFRSF21* transcripts, and *circDHTKD1* was back-spliced by exons 2 and 3 of *DHTKD1* transcripts. Meanwhile, the parental genes *SOD2*, *TNFRSF21*, and *DHTKD1* were previously reported to generate other circular RNAs. Exons 1 and 4 of *SOD2* transcripts were back-spliced to develop *circSOD2*, inducing epigenetic alteration and driving hepatocellular carcinoma progression by activating the JAK2/STAT3 signaling pathway [41]. Exons 2 and 3 of *TNFRSF21* transcripts were back-spliced to produce *circTNFRSF21* that can promote endometrial carcinoma pathogenesis by regulating the *miR-1227*-*MAPK13*/*ATF2* axis [42]. Exons 2 and 16 of *DHTKD1* transcripts were back-spliced to develop *circDHTKD1* that can promote lymphatic metastasis of bladder cancer by upregulating *CXCL5* [43]. In conjunction with our research, it is once again demonstrated that the back-splicing of parental genes can form different functional circRNAs. Meanwhile, we suggest that *circSOD2*, *circCHSY1*, *circTNFRSF21*, and *circDHTKD1* play a critical role in *M.tb* infection.

### 3.3. Predicted Interaction Network of Novel Identified circRNAs in M.tb-Infected Cells

CircRNAs primarily function as super sponges in the cytoplasm, modulating the expression of miRNA targets [11,27,28]. We further conducted nucleocytoplasmic isolation qPCR and found that *circSOD2*, *circCHSY1*, *circTNFRSF21*, and *circDHTKD1* might be present in the cytoplasm of *M.tb*-infected THP-1 cells. Therefore, we constructed an *M.tb* infection-related circRNA-miRNA-mRNA ceRNA network based on these validated circRNAs and their bioinformatic-predicted targets. We matched all the target mRNAs of miRNAs to the tuberculosis signaling pathway ([*hsa-05152*]) and found that 47 target mRNAs were enriched in this pathway. Among these prediction RNAs, the high expression levels of *miR-660* may activate the AKT/NF-κB signaling pathway and have the potential to serve as a biomarker for the diagnosis of pulmonary tuberculosis [44]. *IL-12* and *IL-23* might play a role in supporting *IFN-γ*–mediated protection against mycobacterial infections [45]. *MyD88* signaling fosters bacterial containment and is essential to induce an adequate innate and acquired immune response to *M.tb* infection [46]. In the monocytes of patients with severe tuberculosis, *IL17A* could not augment autophagy because of a defect in the MAPK1/3 signaling pathway [47]. This suggests that the development of the *circDHTKD1*/*miR-660-3p*/*IL-12B* axis, *circTNFRSF21*/*miR-6721-5p*/*MYD88* axis, and *circSOD2*/*miR-4519*/*MAPK1* axis has potential associations with *M.tb* infection. Overall, our findings predicted the potential roles of these circRNAs in the pathogenesis of tuberculosis and anti-TB defense of host cells. Further research is necessary to confirm these findings and elucidate the specific mechanisms involved in the function of circRNAs.

### 3.4. Limitation

Although we conducted a systematic analysis of circRNAs in THP-1 macrophages that were uninfected and infected with virulent and avirulent *M.tb* strains, only 40 circRNAs were confirmed and four were further investigated. Therefore, most of them remain to be explored in the future. Moreover, only 45.02% of sequenced circRNAs were annotated in the available database, and still the lack of data from primary macrophages and in vivo studies limits a comprehensive understanding of the entire picture of circRNA differential profiles induced by *M.tb* infection.

## 4. Materials and Methods

### 4.1. Bacterial Strains and Cell Culture

The *M.tb* strains and infected cells were cultured at the facility in an animal biosafety level-3 (ABSL-3) in Huazhong Agricultural University, Wuhan, China. The virulent *M.tb* 1458 strain (virulent *M.tb*) (GenBank accession no. CP013475.1) was a typical Beijing family strain that isolated from a diseased cattle with TB in this Lab [48,49]. The avirulent H37Ra (H37Ra) was purchased from ATTCC (# 25177) and preserved by this lab. These *M.tb* strains were grown in Middlebrook 7H9 broth medium (BD PharMingen, San Diego, CA, USA), supplemented with 10% oleic acid, albumin, dextrose, and catalase medium (OADC; BD PharMingen), as well as 0.5% glycerol (Sigma-Aldrich, St. Louis, MO, USA) and 0.05% Tween 80 (Sigma-Aldrich).

The THP-1 human monocytic cell line was purchased from ATTCC (# TIB-202) and maintained in RPMI-1640 medium (HyClone, Logan, UT, USA) supplemented with 10% FBS (FBS, Gibco, Grand Island, NY, USA) at 37 °C with 5% CO_2_. Before infection, the THP-1 cells were stimulated with 40 ng/mL phorbol 12-myristate 13-acetate (PMA, Sigma-Aldrich, St. Louis, MO, USA) for 12 h.

### 4.2. Cell Infection with Bacteria

PMA-differentiated THP-1 cells (1 × 10^6^ cells/well) were seeded on a 12-well plate and infected with *M.tb* 1458 or H37Ra at an MOI of 10 as described previously [50] and then incubated for 12 h (defined as −12 h), followed by thorough washing with fresh medium containing 10% FBS and 100 μg/mL gentamycin. This time point was set as the starting point (0 h). Then, the cell culture was continued in complete medium containing 100 μg/mL gentamicin (Sigma-Aldrich) for 12 h, after which the cells were collected for further analysis. All the experiments were performed in triplicate, and at 12 h post infection (PI), the condition of the THP-1 cells and the quantity of intracellular *M.tb* 1458 or H37Ra in the THP-1 cells was confirmed by using an acid-fast staining kit (Yuan Ye Biological Technology, Shanghai, China) and microscopy according to the manufacturer’s instruction (Appendix A).

### 4.3. CircRNA Sequencing and Differential Expression Profile Analysis

Total RNA extraction and circRNA sequencing (GSE 248244) were performed by GENESEED Biotech (Guangzhou, China). Briefly, RNA was harvested using Magen Hipure Total RNA Mini Kit (Magen Biotechnology Co., Ltd., Guangzhou, China). Analysis of the quality and integrity of the RNA used agarose gel electrophoresis, Q30 quality control, and Agilent 2100 Bioanalyzer (Agilent Technologies, Santa Clara, CA, USA). Then, 2 μg of the total RNA was treated with RNase R enzyme (Epicentre, Inc., Madison, WI, USA) to enrich circRNAs for the construction of sequencing libraries and double-ended sequencing was performed on an Illumina HiSeq X10 PE150 (Illumina, Inc., San Diego, CA, USA). The cRNA was labeled with fluorescence using the Arraystar Super RNA Labeling protocol and random primers. The reads were first mapped to the latest UCSC transcript set using Bowtie2 version 2.1.0 [51], and the gene expression level was quantified using RSEM v1.2.15 [52]. Subsequently, the expression data were normalized using TMM (trimmed mean of M-values) and differentially expressed genes were identified using edgeR [53]. A threshold of *p* < 0.05 and fold change > 1.5 was applied to select the genes with significant expression changes.

For the circRNA expression analysis, the reads were aligned to the genome using STAR [54] and the circRNAs and their expression levels were detected using DCC [55]. edgeR [53] was used to identify differentially expressed circRNAs, while the R clusterprofiler package [56] was used to visualization of the differential circRNA expression profile. The R circlize package [57] was used to screening and plotting of differential circular RNAs. Further, the miRanda (https://www.bioinformatics.com.cn (accessed on 10 August 2019)) was used to predict the miRNA targets of the circRNAs. The R software (version 3.6.1) package was used for quantile normalization and data processing [58]. Statistical differences were assessed using Student’s *t*-test, and the *p*-value was adjusted using a false discovery rate (FDR). CircRNAs with an FDR < 0.05 and a fold change ≥ 1.5 were considered differentially expressed between groups.

### 4.4. RNase R Digestion Analyses

To enrich for circRNAs and remove linear RNAs, RNase R digestion was performed on the RNA samples. Each sample (0.5–1 μg) was resuspended in 52 μL of DEPC water and split into two aliquots, one for RNase R treatment and another for control. The RNase R group was incubated with 3 μL of 10× RNase R Reaction Buffer and 1 μL RNase R (20 U/μL), whereas the control group was incubated with 1 μL of DEPC water, at 37 °C for 1–2 h. Digestion was stopped by adding 30 μL of phenol-chloroform-isoamyl alcohol and centrifuging at 13,000× *g* for 5 min at 4 °C. The supernatant was discarded, and the RNA pellet was resuspended with 6 μL of 4 M lithium chloride, 1 μL of glycogen, and 90 μL of precooled absolute ethanol. The precipitate was stored at −80 °C for 1 h and then centrifuged at 13,000× *g* for 20 min at 4 °C, followed by washing twice with 75% precooled ethanol and air-drying. The final RNA pellet was dissolved in 20 μL of DEPC water (Sigma-Aldrich), and the RNase R-enriched RNA was subjected to further experimental analyses. All experiments were performed in triplicate.

### 4.5. Reverse Transcription Polymerase Chain Reaction (RT-PCR)

Specific divergent primers and convergent primers were designed for the predicted circRNAs (Table 1). Total RNA was extracted as described previously [50]. The HiScript^®^ Ⅱ Q RT SuperMix (Vazyme, Nanjing, China) was used to synthesize cDNA from 500 ng of total RNA for analyzing the circRNA expression according to the manufacturer’s instructions. The gene-specific primers (Tsingke Biotech Co., Ltd., Beijing, China) used for PCR are listed in Table 1. The PCR conditions were as follows: initial denaturation at 95 °C for 30 s, 40 cycles of denaturation at 95 °C for 5 s, and annealing at 60 °C for 30 s. All experiments were performed in triplicate.

### 4.6. Quantitative Real-Time PCR (qPCR)

The circRNA expression was normalized to that in the uninfected control, which was set as 1. Subsequently, qPCR analysis was performed using an ABI ViiA 7 Real-Time PCR System (Applied Biosystems, Carlsbad, CA, USA) and the AceQ qPCR SYBR Green master mix (Vazyme). The relative expression of circRNA was normalized using *β-actin* [59]. The gene-specific primers (Tsingke Biotech Co., Ltd.) used for qPCR are listed in Table 1. Data were expressed using 2^−ΔΔCt^ [60], and all experiments were performed in triplicate.

### 4.7. Nucleocytoplasmic Isolation

Subcellular fractionation was performed to determine the cellular localization of circRNAs, for which cytoplasmic and nuclear RNAs were isolated using PAKIS Kit (Thermo Fisher Scientific, Waltham, MA, USA) according to the manufacturer’s instructions. Then, 500 ng of the cytoplasmic or nuclear RNAs were reverse-transcribed using the HiScript^®^ Ⅱ Q RT SuperMix (Vazyme) at 50 °C for 15 min, and 85 °C for 5 s, and the expression of genes was measured by qPCR analysis. *NEAT1* was used as the nuclear control, and *GAPDH* was used as the cytoplasmic control. The specific primers (Tsingke Biotech Co., Ltd.) are listed in Table 1, and all experiments were performed in triplicate.

### 4.8. KEGG and Reactome Enrichment Analyses

KEGG pathway analyses were conducted using the R cluster profiler package [61] and R package networkD3 [62] that use hypergeometric distribution to perform functional classification and enrichment of gene clusters.

### 4.9. Prediction of miRNA-circRNA and circRNA-miRNA-circRNA ceRNA Network

StarBase v2.0 was used to predict the downstream miRNAs of the selected circRNAs. Data were visualized using the networkD3 R package [62]. To gain more insights from bioinformatics tools, we used the following three prediction algorithms to identify the potential target mRNAs of the target miRNAs of circRNAs: microRNA (http://www.microrna.org (accessed on 15 June 2020)); Targetscans7.1 (http://www.targetscan.org (accessed on 15 June 2020)); and miRBD (http://mirdb.org/miRDB/ (accessed on 15 June 2020)). These algorithms evaluated the complementarity, conservation, and thermodynamic stability between the 3′-UTR of mRNA and the 5′seed region of miRNAs. The genes predicted by all three algorithms were considered candidate targets of the miRNAs. The Sankey plots demonstrated the predicted circRNA-miRNA-circRNA interaction network results for each circRNA.

### 4.10. Statistical Analysis

Data are expressed as mean ± SD (standard deviation) of three biological replicates for each sample. GraphPad Prism software (Version 7.0, Boston, MA, USA) was used for statistical analysis and significance was determined using an unpaired two-tailed Student’s *t*-test for one comparison and analysis of variance (ANOVA) for multiple comparisons, as appropriate. Differences were considered statistically significant when *p* < 0.05, and *p* values of <0.05, <0.01, and <0.001 were respectively indicated as *, **, and *** in figures.

## 5. Conclusions

Using circRNA sequencing and bioinformatic tools, we identified 58,009 circRNAs in the *M.tb*-infected and uninfected THP-1 macrophages. A total of 2035 differentially expressed circRNAs were identified between the *M.tb*-infected and uninfected THP-1 cells, and 1258 circRNAs were identified between the virulent and avirulent *M.tb* strains. Importantly, we discovered four novel circRNAs—*circSOD2*, *circCHSY1*, *circTNFRSF21*, and *circDHTKD1* that could be used as potential diagnostic markers and therapeutic targets for *mycobacterium* infection. We also constructed a network of interactions among these four circRNAs, their target miRNAs, and their target mRNAs. These networks might provide a novel perspective to understand the possible roles of circRNAs in the interaction between the *M.tb* and macrophages.

## Figures and Tables

**Figure 1 ijms-24-17561-f001:**
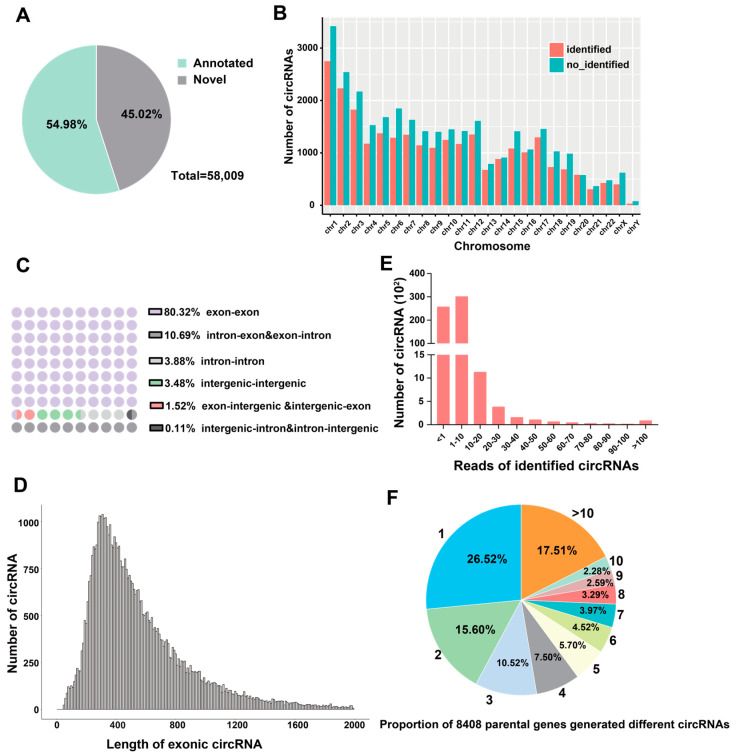
Characterization of identified circRNAs in *M.tb*-infected and uninfected THP-1 macrophages. (**A**) The proportion of annotated circRNAs in CircBase and novel circRNAs identified in our samples. (**B**) The chromosome distribution of all circRNAs in macrophages. (**C**) Pie graph depicting the types and percentages of circRNAs. (**D**) The length distribution of circRNAs. (**E**) The number of circRNA and average back-splicing reads in this study. (**F**) The proportions of 8408 parental genes that generated different circRNAs. Numbers outside the circle indicate the number of circRNAs generated by one host gene, and percentages indicate the proportions of parental genes among the 8408 genes.

**Figure 2 ijms-24-17561-f002:**
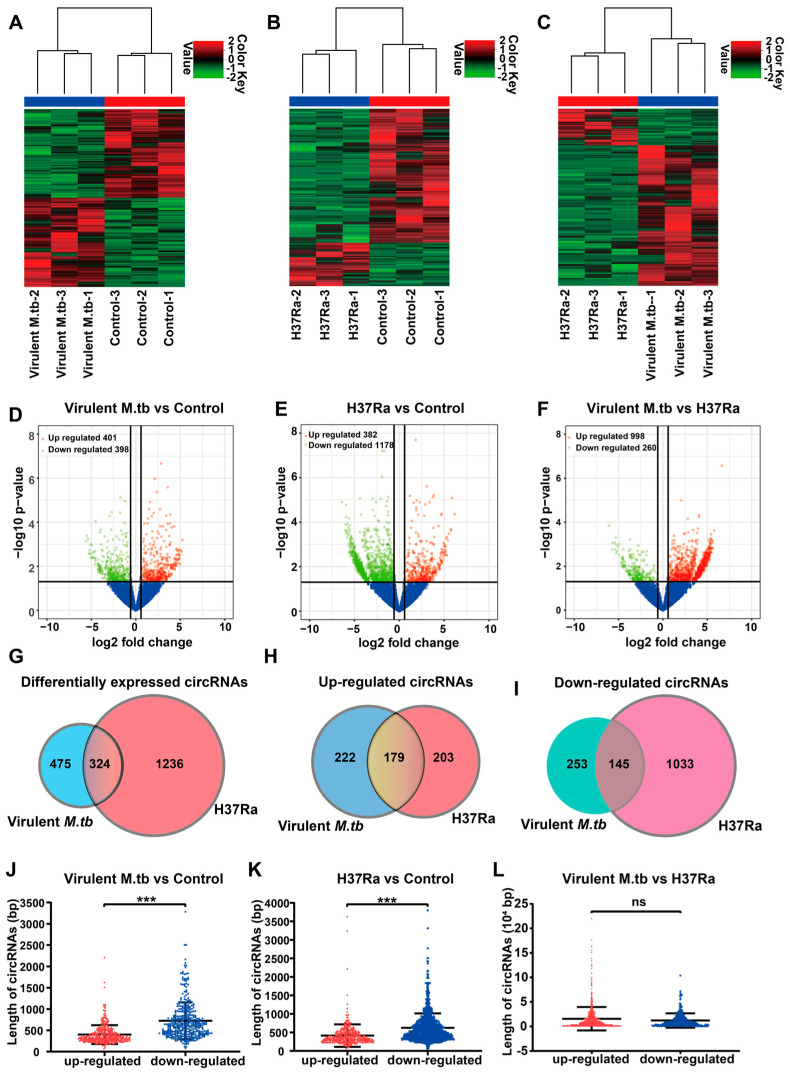
Differentially expressed circRNAs between infected and uninfected THP-1 macrophages. (**A**) Hierarchical clustering heatmap of differentially expressed circRNAs between virulent *M.tb*-infected macrophages and uninfected controls. (**B**) Hierarchical clustering heatmap of differentially expressed circRNAs between H37Ra-infected macrophages and uninfected controls. (**C**) Hierarchical clustering heatmap of differentially expressed circRNAs between virulent *M.tb*- and H37Ra-infected macrophages. (**D**) The not differentially expressed, significantly upregulated and downregulated circRNAs in virulent *M.tb*-infected macrophages and controls, which are colored blue, red and green, respectively. The vertical line represents 1.5-fold upregulation and downregulation, and the horizontal line corresponds to *p* = 0.05. (**E**) The not differentially expressed, significantly upregulated and downregulated circRNAs in H37Ra-infected macrophages and controls, which are colored blue, red and green, respectively. The vertical line represents 1.5-fold upregulation and downregulation, and the horizontal line corresponds to *p* = 0.05. (**F**) The not differentially expressed, significantly upregulated and downregulated circRNAs in virulent *M.tb*- and H37Ra-infected macrophages, which are colored blue, red and green, respectively. The vertical line represents 1.5-fold upregulation and downregulation, and the horizontal line corresponds to *p* = 0.05. (**G**) Venn plot of differentially expressed circRNAs in virulent *M.tb*-infected macrophages between the differentially expressed circRNAs in H37Ra-infected macrophages. (**H**) Venn plot of upregulated circRNAs in virulent *M.tb*-infected macrophages between the upregulated circRNAs in H37Ra-infected macrophages. (**I**) Venn plot of downregulated circRNAs in virulent *M.tb*-infected macrophages between the downregulated circRNAs in H37Ra-infected macrophages. (**J**) Comparison of the length of upregulated and downregulated circRNAs in *M.tb*-infected macrophages with those in controls. (**K**) Comparison of the length of upregulated and downregulated circRNAs in H37Ra-infected macrophages with that in uninfected controls. (**L**) Comparison of the length of upregulated and downregulated circRNAs in virulent *M.tb*- and H37Ra-infected macrophages. *p* < 0.001 was presented by ***. Abbreviations: virulent *M.tb*, virulent *M.tb* 1458 strain; H37Ra, avirulent *M.tb* H37Ra strain. ns: not significance.

**Figure 3 ijms-24-17561-f003:**
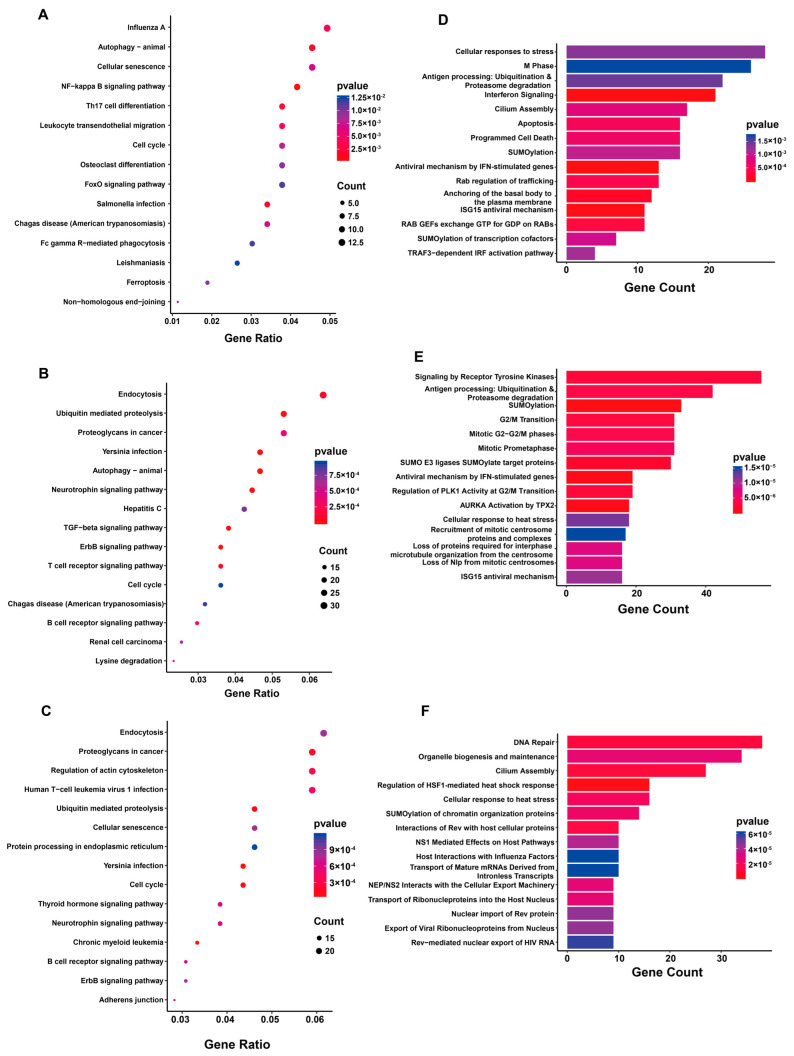
KEGG and Reactome pathway analysis of differentially expressed circRNA parental genes. (**A**–**C**) KEGG pathway analysis of circRNA parental genes in virulent *M.tb*-infected macrophages or/and H37Ra-infected macrophages. The x-axis represents the gene ratio, which is the ratio of the number of intersecting genes between the differentially expressed gene set and the KEGG term to the total number of differentially expressed genes. The y-axis represents the KEGG term. The size of the bubble indicates the number of intersecting genes. (**D**–**F**) Reactome pathway analysis of differentially expressed circRNA parental genes between virulent *M.tb*-infected and H37Ra-infected macrophages. The x-axis represents the gene count, and the y-axis represents the KEGG term. Abbreviations: virulent *M.tb*, virulent *M.tb* 1458 strain; H37Ra, avirulent *M.tb* H37Ra strain.

**Figure 4 ijms-24-17561-f004:**
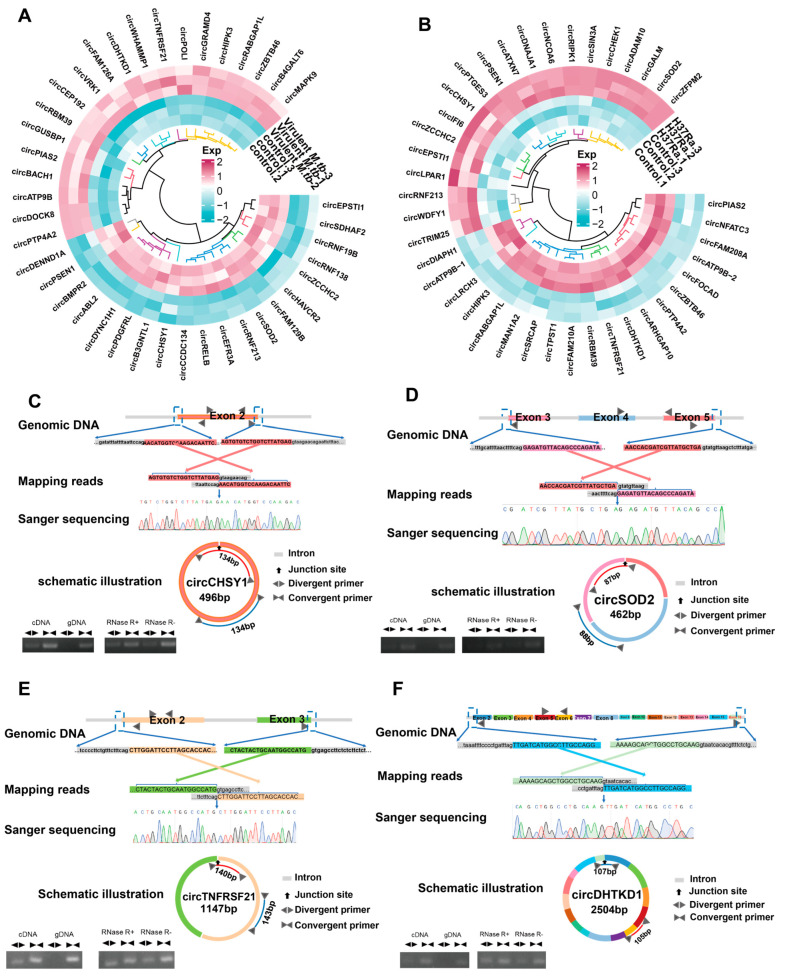
Selection and detection of differentially expressed circRNAs. (**A**) Top 40 differentially expressed circRNAs in virulent *M.tb*-infected macrophages. (**B**) Top 40 differentially expressed circRNAs in H37Ra-infected macrophages. (**C**–**F**) Validating the back-spliced reads of selected circRNAs by multiple methods, including agarose electrophoresis and Sanger sequencing of RT-PCR products amplified using specially designed divergent primers and convergent primers. The diagrams and details of the four circRNAs (*circCHSY1*, *circSOD2*, *circTNFRSF21*, and *circDHTKD1*) selected in our study and the primer design and the results of the RNase R digestion experiment, which was used to investigate the stability and resistance of circRNAs to RNase R, are also shown. Moreover, PCR in cDNA and gDNA was performed using convergent primers and divergent primers to validate the origins of the back-splicing reads of circRNAs. Abbreviations: virulent *M.tb*, virulent *M.tb* 1458 strain; H37Ra, avirulent *M.tb* H37Ra strain.

**Figure 5 ijms-24-17561-f005:**
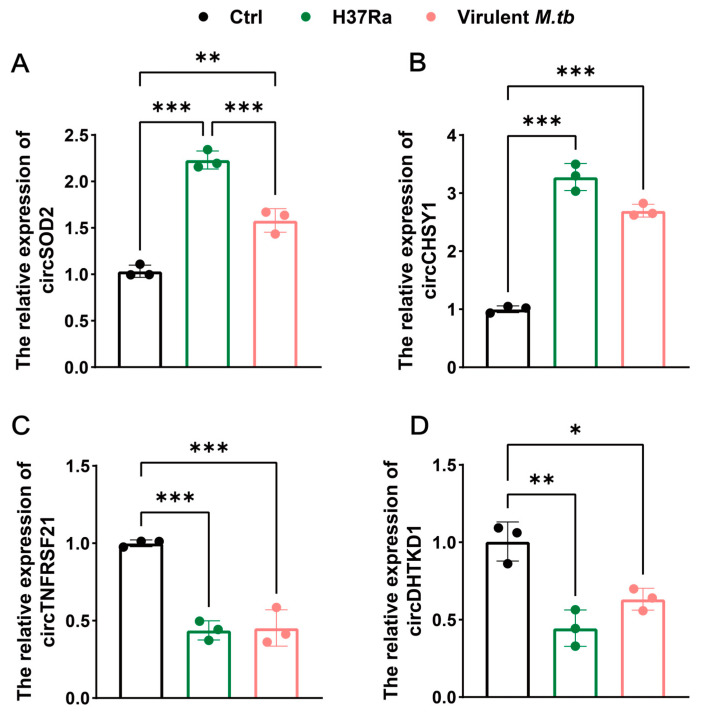
Detection of selected differentially expressed circRNAs by RT-qPCR. THP-1 macrophages that were infected with *M.tb* at an MOI of 10 for 12 h, and the expressions of four selected circRNAs, including (**A**) *circSOD2*, (**B**) *circCHSY1*, (**C**) *circTNFRSF21*, and (**D**) *circDHTKD1*, in virulent *M.tb*- and H37Ra-infected macrophages were detected by RT-qPCR. Data represent one of three experiments. *p* < 0.05, <0.01, and <0.001 are presented by *, **, and ***, respectively. Abbreviations: virulent *M.tb*, virulent *M.tb* 1458 strain; H37Ra, avirulent *M.tb* H37Ra strain.

**Figure 6 ijms-24-17561-f006:**
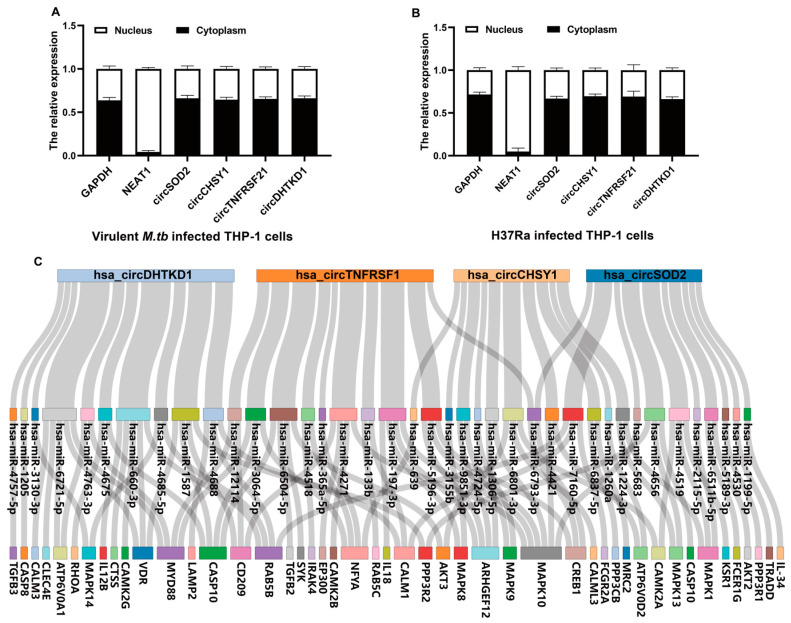
Prediction of *M.tb* infection-related circRNA-miRNA-mRNA competing endogenous interaction network. (**A**,**B**) Expression levels of *circSOD2*, *circCHSY1*, *circTNFRSF21*, and *circDHTKD1* in the cytoplasm and nucleus in virulent *M.tb*- (**A**) and H37Ra (**B**)-infected THP-1 cells. Cytoplasmic and nuclear RNAs of THP-1 cells were isolated using PAKIS Kit, reverse-transcribed immediately, the expression of genes was measured by qPCR, and *NEAT1* was used as the nuclear control and *GAPDH* was used as the cytoplasmic control. (**C**) The competing endogenous interaction network is shown in the Sankey diagram, the top 10 miRNAs that potentially combine with one circRNA are shown in the middle, and the four identified circRNAs are shown above the miRNAs. The target genes related to *M.tb* infection of miRNAs are displayed below. The edge size between circRNAs and miRNAs correlated with the number of mRNAs they potentially regulated. Data represent one of three experiments. Abbreviations: virulent *M.tb*, virulent *M.tb* 1458 strain; H37Ra, avirulent *M.tb* H37Ra strain.

**Table 1 ijms-24-17561-t001:** Primers used for RT-PCR and qPCR.

Primer Names	Sequences (5′–3′)	Products (bp)
*hsa-β-actin*	F: CATGTACGTTGCTATCCAGGC R: CTCCTTAATGTCACGCACGAT	250
*hsa-NEAT1*	F: AAACGCTGGGAGGGTACAAG R: ATGCCCAAACTAGACCTGCC	71
*hsa-GAPDH*	F: AATGGGCAGCCGTTAGGAAA R: GCCCAATACGACCAAATCAGAG	166
*hsa-circCHSY1*- Divergent	F: AAGGTGTGTCCGGAGGTTTG R: TGGCACTACTGGAATTGGTACA	134
*hsa-circCHSY1*- Convergent	F: TGCACGACCACTACTTGGAC R: GCCTGTCTGCCCAAGAAAGA	134
*hsa-circSOD2*- Divergent	F: AATGTAATCAACTGGGAGAATG R: GGCTGTAACATCTCTCAGCATA	87
*hsa-circSOD2*- Convergent	F: CTGGAAGCCATCAAACGTGAC R: AACCTGAGCCTTGGACACC	88
*hsa-circTNFRSF21*- Divergent	F: TACTGCAATGGCCATGCTTG R: TTCCTGCTGGACACTTGTCA	140
*hsa-circTNFRSF21*- Convergent	F: CTGCCTTGACTGACCGAGAA R: ACACTGCTTACACCGCACAT	143
*hsa-circDHTKD1*- Divergent	F: CGTGGTCGTTTGTTTCTCCA R: TGGCAGCTTTATGACCATGC	107
*hsa-circDHTKD1*- Convergent	F: TGCTTACAGGTCCATGGTGA R: ATGCACACTCCCACCAATTC	105

## Data Availability

The datasets generated in this study are available upon request from the corresponding authors.

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
