# Peer review of "Identification of Differential Circular RNA Expression Profiles and Functional Networks in Human Macrophages Induced by Virulent and Avirulent Mycobacterium tuberculosis Strains"

_ijms, 2023, doi:10.3390/ijms242417561_

Round 1

Reviewer 1 Report

Comments and Suggestions for Authors

In this article, Zhu et.al used THP-1-derived macrophages to determine the host circular RNA expression profile after infection with pathogenic and non-pathogenic mycobacteria.  The authors have used bioinformatics approach combined with qPCR and molecular assays to characterize 4 circular RNAs that may be relevant to mycobacterial infection of the THP-1 macrophages.  This is a simple and straight-forward experimental study that is well-presented.  However, there are some issues that needs to be addressed:

The writings on figures are too small to read, especially Figure 2 (A, B, C), Figure 3 and Figure 4 that even when magnified to >200x, the lettering on the x and y axis are not readable. These writings, including the scale bar markings must be changed to improve clarity.

Describe gene ratio in the legend of Figure 3.

Line 210. It is unclear why and how those 4 circRNAs (CHSY1, SOD2, TNFRSF21 and DHTKD11) were selected for further analysis. Please provide the rationale for the selection.  In addition, it would be more valuable if the authors have examined few of the top up/down molecules that are unique to pathogenic or non-pathogenic Mtb strains, which would be a better validation of biomarker and make more sense, considering the comparison in Figure 1A-C.

From Figures 5B, C, D and 6A,B it is clear that there was no significant difference in the expression pattern of circCHSY1, TNFRSF21 and DHTKD1 between virulent and avirulent strains of mycobacterium.  Therefore, the authors should be careful in interpreting the data. By using a single abbreviation of M. tb, the authors underestimate the distinction between virulent and avirulent M. tb strains.  I strongly recommend the authors to use different abbreviations (e.g, Rv and Ra) to clarify what are the factors relevant to a pathogen (Rv) and/or to a non-pathogen (Ra).

Line 377. In the limitation section, include the lack of any data from primary macrophages and in vivo studies.

Line 387-389.  Mention the source of these Mtb strains (i.e, where was it obtained from, and who was the supplier)?. Just mentioning the genome accession no. of these strains is not enough.

Line 391.  Mention the source of the THP-1 cells (i.e, where was it obtained from, and who was the supplier)?

Line 397. Mention the experimental design in brief (number of cells, well type, how many wells/group etc). Did the authors measure the viability of THP-1 cells before and after infection with Mtb infection? This information should be mentioned in line 402.

Line 405. Cite a reference for Geneseed procedure and briefly mention it here (the extraction method, quality/quantity check, and how much of RNA was used etc).

Line 436. Briefly mention the experimental setup (how much RNA and what buffer/kit was used).

Line 452. Briefly mention the experimental setup and how much RNA was used for RT reaction?

Line 472. The statistical analysis section is too vague. The authors must explicitly mention which data were analyzed by T-test and which ones by one-way ANOVA?

Line 473. What software used for statistical analysis? 

Line 482. Change the wordings, because the selected markers are not exclusive to pathogenic Mtb strain (see figures 5 and 6).

It is very important that the number of biological and technical replicates used for each of the experiment must be explicitly mentioned in the methods section and in the legends.

Reviewer 2 Report

Comments and Suggestions for Authors

The manuscript titled "Identification of differential circRNA expression profiles and functional networks in human macrophages induced by virulent and avirulent Mycobacterium tuberculosis strains" provides clues about Mycobacterium tuberculosis infection by deriving meaningful circRNAs based on a review of the crucial functions of circular RNA in the body. While the experimental data are supported by relatively credible experimental evidence, for the paper to meet the quality required for acceptance in IJMS, more background knowledge needs to be provided, and experimental data that can satisfy the conclusions need to be further interpreted.

* Major Points:

1. The paper focuses on the perspective of macrophage infection. However, it should also explain cases of infection with different types in the introduction part.

2. The process of deriving significant circular RNA is presented merely as numbers. The authors need to provide a more systematic methodology by explaining the algorithm they used.

3. The term 'validation' is used in Result 2.4, but if actual pathogenic strains were not used, accurate validation cannot be claimed. The authors should more clearly argue that the results are supported even with non-pathogenic strains like Mycobacterium smegmatis, which is almost identical to the tuberculosis bacterium.

4. Figure 6c covers a broad scope. The coverage should be reduced, and the presented parts need more detailed explanations. If the current format is maintained and some content cannot be explained, it should be excluded, or supplementary data should be attached if the content is too extensive.

* Minor Point:

5. Remove the reference panel in Table 1 and add the corresponding reference to the table legend. This table composition is not suitable.

Round 2

Reviewer 2 Report

Comments and Suggestions for Authors

The authors have responded diligently to my inquiries, and the quality of the manuscript has significantly improved. Now, this paper is ready for acceptance in IJMS. Congratulations!